# Quality of care of peptic ulcer disease worldwide: A systematic analysis for the global burden of disease study 1990–2019

Mohsen Abbasi-Kangevari[1‡], Naser Ahmadi[1‡], Nima Fattahi[1], Negar Rezaei[1,2], Mohammad-Reza Malekpour[1], Seyyed-Hadi Ghamari[1], Sahar Saeedi Moghaddam[1], Sina Azadnajafabad[1], Zahra Esfahani[1,3], Ali-Asghar Kolahi[4], Shahin Roshani[1,5], Sahba Rezazadeh-Khadem[1], Fateme Gorgani[1], Seyyed Nima Naleini[6], Shohreh Naderimagham[1,7], Bagher Larijani[2], Farshad Farzadfar[1,2]*

1 Non-Communicable Diseases Research Center, Endocrinology and Metabolism Population Sciences Institute, Tehran University of Medical Sciences, Tehran, Iran, 2 Endocrinology and Metabolism Research Center, Endocrinology and Metabolism Clinical Sciences Institute, Tehran University of Medical Sciences, Tehran, Iran, 3 Department of Biostatistics, University of Social Welfare and Rehabilitation Sciences, Tehran, Iran, 4 Social Determinants of Health Research Center, Shahid Beheshti University of Medical Sciences, Tehran, Iran, 5 The Netherlands Cancer Institute(NKI), Amsterdam, Netherlands, 6 Student Research Committee, Kurdistan University of Medical Sciences, Sanandaj, Iran, 7 Elderly Health Research Center, Endocrinology and Metabolism Population Sciences Institute, Tehran University of Medical Sciences, Tehran, Iran

‡ MAK and NA share co-first author on this work.
* f-farzadfar@tums.ac.ir

**Data Availability Statement:** The study protocol and codes used in this study are available from (https://dx.doi.org/10.17504/protocols.io.

## Abstract

### Background

Peptic ulcer disease (PUD) affects four million people worldwide annually and has an estimated lifetime prevalence of 5−10% in the general population. Worldwide, there are significant heterogeneities in coping approaches of healthcare systems with PUD in prevention, diagnosis, treatment, and follow-up. Quantifying and benchmarking health systems' performance is crucial yet challenging to provide a clearer picture of the potential global inequities in the quality of care.

### Objective

The objective of this study was to compare the health-system quality-of-care and inequities for PUD among age groups and sexes worldwide.

### Methods

Data were derived from the Global Burden of Disease Study 1990–2019. Principal-Component-Analysis was used to combine age-standardized mortality-to-incidence-ratio, disability-adjusted-life-years-to-prevalence-ratio, prevalence-to-incidence-ratio, and years-of-life-lost-to-years-lived-with-disability-into a single proxy named Quality-of-Care-Index (QCI). QCI was used to compare the quality of care among countries. QCI's validity was investigated via correlation with the cause-specific Healthcare-Access-and-Quality-index, which

bprjmm4n) [5]. The data used in this work are available from Global Burden of Disease Results Tool (http://ghdx.healthdata.org/gbd-results-tool) [6], made public by Institute for Health Metrics and Evaluation. The data of PUD were extracted from GBD 2019: GBD code: B.4.2.1, International Statistical Classification of Diseases and Related Health Problems 10th Revision, World Health Organization version 10 (ICD-10) code: K-25 to K28.9 [7]. Data sources used to provide estimates in GBD 2019 are presented in S1 Table. In terms of the development status, countries were categorized using the GBD Socio-Demographic Index (SDI) [8].

**Funding:** The author(s) received no specific funding for this work.

**Competing interests:** The authors have declared that no competing interests exist.

was acceptable. Inequities were presented among age groups and sexes. Gender Disparity Ratio was obtained by dividing the score of women by that of men.

## Results

Global QCI was 72.6 in 1990, which increased by 14.6% to 83.2 in 2019. High-income-Asia-pacific had the highest QCI, while Central Latin America had the lowest. QCI of high-SDI countries was 82.9 in 1990, which increased to 92.9 in 2019. The QCI of low-SDI countries was 65.0 in 1990, which increased to 76.9 in 2019. There was heterogeneity among the QCI-level of countries with the same SDI level. QCI typically decreased as people aged; however, this gap was more significant among low-SDI countries. The global Gender Disparity Ratio was close to one and ranged from 0.97 to 1.03 in 100 of 204 countries.

## Conclusion

QCI of PUD improved dramatically during 1990–2019 worldwide. There are still significant heterogeneities among countries on different and similar SDI levels.

## Introduction

Peptic ulcer disease (PUD) affects four million people worldwide annually [1] and has an estimated lifetime prevalence of 5–10% in the general population [2]. Although the global prevalence of PUD has dramatically decreased in the past decades [3], the incidence of its complications has remained constant [4].

Worldwide, there are significant heterogeneities in coping approaches of healthcare systems with PUD in terms of prevention, diagnosis, treatment, and follow-up [5]. Prevention is positively correlated with the development of infrastructures and the effectiveness of healthcare systems [6]. The choice of diagnostic test and treatment approaches mainly relies on accessibility and cost [7]. Therefore, quantifying and benchmarking health systems' performance is crucial yet challenging to provide a clearer picture of the potential global inequities in the quality of care [8].

In this sense, the objective of this study was to compare the quality of medical care provided for PUD using the Quality of Care Index (QCI) among age groups and both sexes in various nations and regions based on the data of the Global Burden of Disease (GBD) Study 2019. To obtain QCI, four indices of PUD, including age-standardized mortality to incidence ratio, disability-adjusted life years (DALYs) to prevalence ratio, prevalence to incidence ratio, and years of life lost (YLLs) to years lived with disability (YLDs) ratio, were combined into a single index using Principal Component Analysis (PCA) [9–17].

## Materials and methods

The study data were initially derived from the GBD study, conducted by Institute for Health Metrics and Evaluation (IHME), which abides by relevant guidelines and regulations. We carried out no experiments on any human or animal subjects. Thus, consent to participate does not apply to the present study. This study was approved by the institutional review board of Endocrinology and Metabolism Research Institute at Tehran University of Medical Sciences (IR.TUMS.EMRI.REC.1400.016).

## Data source

The study data were derived from GBD 2019, conducted by IHME. GBD 2019 included 204 countries and territories from 1990 to 2019 and a systematic analysis of 286 causes of death, 369 diseases and injuries, and 87 risk factors in 204 countries and territories [18, 19]. GBD classified countries and territories into 21 regions based on epidemiological homogeneity and geographical contiguity [20]. The regions were also grouped into seven super-regions based on the cause of death patterns [21]. The seven super-regions are High income; Latin America & Caribbean; Sub-Saharan Africa; North Africa & Middle East; Southeast Asia, East Asia & Oceania; South Asia; Central Europe, Eastern Europe & Central Asia.

The study protocol and codes used in this study are available from (https://dx.doi.org/10.17504/protocols.io.bprjmm4n) [22]. The data used in this work are available from Global Burden of Disease Results Tool (http://ghdx.healthdata.org/gbd-results-tool) [23], made public by Institute for Health Metrics and Evaluation. The data of PUD were extracted from GBD 2019: GBD code: B.4.2.1, International Statistical Classification of Diseases and Related Health Problems 10th Revision, World Health Organization version 10 (ICD-10) code: K-25 to K28.9 [24]. Data sources used to provide estimates in GBD 2019 are presented in S1 Table. In terms of the development status, countries were categorized using the GBD Socio-Demographic Index (SDI) [25].

## Statistical analysis

**Age standardization.** To account for the change in population structure, age-standardized incidence, mortality, YLLs, YLDs, and DALYs rates were computed by direct standardization to the global age structure and expressed as the number per 100,000 population [18]. Nevertheless, to investigate the inequities among age groups, all-ages measures were used to calculate QCI.

**Quality of care index.** To determine QCI for PUD, the following indices were defined as follows:

$$\text{Mortality to incidence ratio of PUD} = \frac{\#Age-standardizaed\ PUD\ mortality}{\#Age-standardizaed\ PUD\ incidence}$$

The mortality to incidence ratio of PUD indicates that with a stable PUD incidence in regions, higher mortality values could represent worse care provided to these patients.

$$\text{DALYs to prevalence ratio of PUD} = \frac{\#Age-standardizaed\ \text{DALYs of PUD}}{\#Age-standardizaed\ PUD\ \text{prevalence}}$$

DALY to prevalence ratio of PUD indicates that with a similar prevalence of PUD in regions, higher DALY could represent worse care quality.

$$\text{Prevalence to incidence ratio of PUD} = \frac{\#Age-standardizaed\ PUD\ \text{prevalence}}{\#Age-standardizaed\ PUD\ incidence}$$

The prevalence to incidence ratio of PUD indicates that in regions with similar PUD incidence, higher prevalence could represent better PUD management to avert mortality.

$$\text{YLLs to YLDs ratio of PUD} = \frac{\#Age-standardizaed\ \text{YLLs of PUD}}{\#Age-standardizaed\ \text{YLDs of PUD}}$$

YLLs to YLDs ratio of PUD could reflect the quality of healthcare in a region, as poor health

quality provided for PUD in a region would result in higher YLLs and fewer YLDs. In other words, patients would live less than the life expectancy of the region.

**Principal components analysis.** PCA would allow us to reduce the number of variables in a large set of correlated variables to a smaller number of variables that collectively explain most of the variance in the original set [26]. The first component of PCA was made up of a linear combination of the four abovementioned ratios, including mortality to incidence, DALYs to prevalence, prevalence to incidence, and YLLs to YLDs, containing the most significant amount of information about these variables and is referred to as QCI in this study. QCI ranged from 0 to 100, with 100 indicating the best quality of care [9–11, 17, 22]. PCA would allow us to reduce the number of variables in a large set of correlated variables to a smaller number of variables that collectively explain most of the variance in the original set The interpretation of all the above measures could be used as a proxy for the quality of care provided for the disease. In this study, PCA was performed to convert the four indices into a single index. PCA is an approach to statistical analysis in which multiple datasets are combined as orthogonal components [26]. The first component of PCA was, made up of a linear combination of the four abovementioned ratios, including mortality to incidence, DALYs to prevalence, prevalence to incidence, and YLLs to YLDs, all variables, containings the greatest most significant amount of information about these variables and is referred to as QCI in this study. QCI ranged from 0 -to 100, with 100 indicating the best quality of care [9–11, 17, 22].

PCA was performed using R software version 3.5.2. For enhanced interpretation and comparison of countries, QCI was categorized as five levels in 2019 based on quintiles, where Level 1 (the first quintile) indicated the highest index and Level 5 (the fifth quintile) the lowest: Level 5 included QCI ≤69.14, Level 4 QCI>69.14 to 75.23, Level 3 QCI>75.23 to 81.33, Level 2 QCI>81.33 to 86.35, and Level 1 QCI>86.35.

**Data validation.** Healthcare Access and Quality (HAQ) was another index for measuring personal healthcare access and quality across countries, developed by GBD in 2016 [27]. HAQ has been calculated for 32 causes, including PUD, that were considered amenable to health care comprise, representing a range of health service areas: vaccine-preventable diseases; infectious diseases and maternal and child health; non-communicable diseases, including cancers, cardiovascular diseases, and other non-communicable diseases such as diabetes; and gastrointestinal conditions from which surgery can avert death. Thus, the cause-specific HAQ reported for PUD could be used to validate QCI externally. The correlation between the QCI and all-causes/cause-specific HAQ index [28, 29] was evaluated by applying a mixed-effect model of QCI as a dependent variable and inpatient and outpatient health care utilization, mortality of PUD, and its mortality attribute to smoking, which is its known risk factor reported by GBD [24], as independent variables while considering countries as random effects. The correlation between the HAQ Index and the predicted values was 0.72, indicating an acceptable association between the two indices [30] and validating QCI as an applicable measure of the quality of care among patients with PUD (S2 Table). Healthcare Access and Quality (HAQ) was another index for measuring personal health-care access and quality across countries, developed by GBD in 2016 [27]. HAQ has been calculated for 32 causes, including PUD, that were considered amenable to health care comprise, representing a range of health service areas: vaccine-preventable diseases; infectious diseases and maternal and child health; non-communicable diseases, including cancers, cardiovascular diseases, and other non-communicable diseases such as diabetes; and gastrointestinal conditions from which surgery can easily avert death. Thus, the cause-specific HAQ reported for PUD could be used to externally validate QCIvalidate QCI externally. The correlation between the QCI and all-causes/cause-specific HAQ index [28, 29] was evaluated by applying a mixed-effect model of QCI as a dependent variable and inpatient and outpatient health care utilization, mortality of PUD, and its mortality

attribute to smoking, which is its known risk factor reported by GBD [24], as independent variables while considering countries as random effects. The correlation between the HAQ Index and the predicted values was 0.72, indicating an acceptable association of between the two indices [30] and validating QCI as an applicable measure of the quality of care among patients with PUD (S2 Table).

**Inequity pattern.** *Geographical inequity.* The age-standardized QCI among various locations were presented as a geographical hierarchy, including 204 countries and territories grouped into 21 regions and seven GBD super-regions [18].

*Age inequity.* Age inequity presented the status of quality of care across different age groups based on different SDI regions. All-ages data were used for determining the QCI among age groups.

*Gender inequity.* The age-standardized QCIs among both women and men were calculated separately and compared to determine the status of gender equity. Gender Disparity Ratio (GDR) was obtained by dividing the age-standardized QCI of women by that of men. The numbers closer to one indicate a better situation.

$$\text{GDR} = \frac{\text{age} - \text{standardized } QCI \text{ for women}}{\text{age} - \text{standardized } QCI \text{ for men}}$$

The statistical analyses were carried out in October 2020 using R statistical packages v3.4.3 (http://www.r-project.org, RRID: SCR_001905). Data visualizations were performed using Python programming language (Python Language Reference, version 3.6. Available at www.python.org) via Altair version 4.1, an open-source Python library. The maps were generated using free open-source map data of Natural Earth public domain (naturalearthdata.com) via Python programming language. The statistical codes are available elsewhere [22].

## Results

### Prevalence, incidence, and mortality

Since 1990, the age-standardized prevalence of PUD has decreased 31% worldwide, from 143.4 per 100,000 to 99.4 in 2019. However, there was heterogeneity in the prevalence changes by geographical distribution. Among 21 GBD regions in 1990–2019, the age-standardized prevalence rate of PUD decreased by almost 70% in Andean Latin America, 58% in Latin America and Caribbean, and 37% in South Asia. PUD incidence has continuously decreased by 31% worldwide, from 63.8 in 1990 per 100,000 to 44.3 in 2019. The age-standardized incidence rate has decreased by almost 68% in Andean Latin America, 56% in Latin America and Caribbean, and 45% in Eastern Europe. The age-standardized mortality rate of PUD has 59% decreased, from 7.4 in 1990 to 3.0 in 2019. The highest age-standardized mortality of PUD was witnessed in low and low-middle SDI countries (S3 and S4 Tables).

### Quality of care index

**GBD regions.** Distinct geographic patterns emerged for QCI levels from 1990 to 2019. The global age-standardized QCI was 72.6 (Level 4) in 1990, which has increased by 14.6% to 83.2 (Level 2) in 2019. Among GBD regions, high-income Asia Pacific had the highest age-standardized QCI in 2019. Central Latin America had the lowest QCI in 1990, which increased by 154.9% from 20.1 (Level 5) to 51.2 (Level 5) in 2019. By 2019, nearly all countries and territories saw increases in their QCI, except for Armenia, Zimbabwe, Democratic People's Republic of Korea, Brazil, Kazakhstan, Uzbekistan, Lesotho, Nepal, Ukraine, Pakistan, Lithuania, Tajikistan, and Paraguay. The top five countries with the highest QCI in 2019 included

Singapore (98.5), Turkey (97.3), Japan (96.9), the Republic of Korea (96.9), and the United States of America (95.1). Honduras (13.2), Guatemala (27.0), Cambodia (40.2), El Salvador (41.1), and Nicaragua (43.5) had the lowest index. The gap between the highest and lowest QCI was narrower in 2019 (ΔQCI = 85.3) than in 1990 (ΔQCI = 89.3). QCI of countries in categories in 1990 and 2019 are presented in Fig 1 and S5 Table.

**QCI levels.** Level 1 included countries from all GBD super-regions: Central Europe, Eastern Europe, and central Asia; High income; Latin America and Caribbean; North Africa and the Middle East; South Asia; Southeast Asia, East Asia, and Oceania; and Sub-Saharan Africa. High income; Latin America and Caribbean; Southeast Asia, East Asia, and Oceania; and Sub-

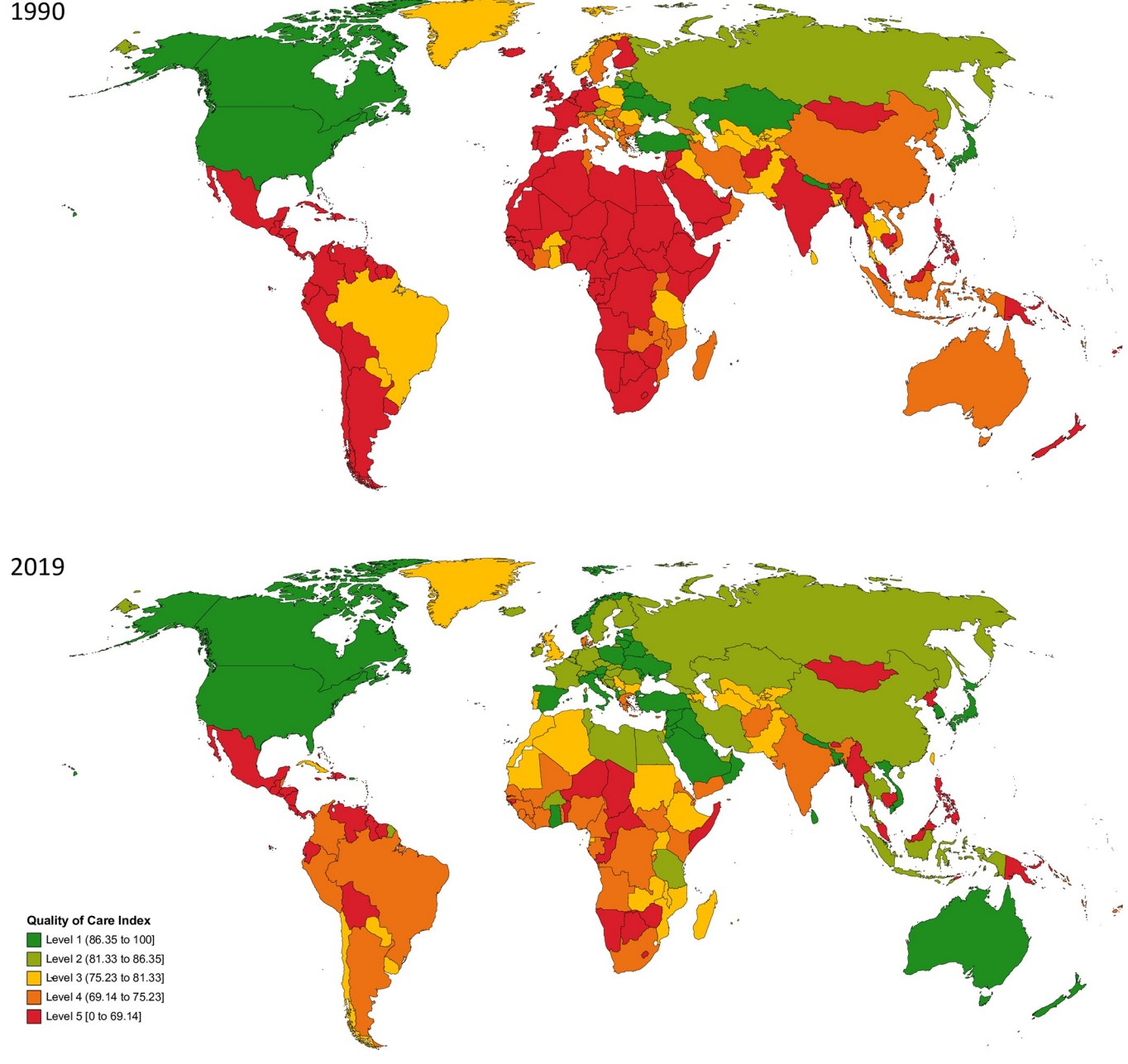

**Fig 1. Quality of care index for peptic ulcer disease by country in 1990 and 2019.**

Saharan Africa had varied age-standardized QCI levels, spanning from Level 5 to Level 1. Among GBD super-regions, South Asia had the most improvement in QCI score (Fig 1 and S5 Table).

**SDI levels.** Age-standardized QCI of high SDI countries was 82.9 (Level 2) in 1990, which increased by 12.0% to 92.9 (Level 1) in 2019. The QCI of low SDI countries was 65.0 (Level 5) in 1990, which increased by 76.9 (Level 2) in 2019. There was heterogeneity among the QCI level of countries with the same SDI level: i.e., the United Kingdom (76.8) and Singapore (98.5). The QCI level of countries with high or high-middle SDI varied from Level 5 to Level 1. No countries with middle, low-middle, or low SDI had a Level 1 QCI score, except for Nepal (Fig 2 and S5 Table).

## Age pattern for quality of care index

QCI typically decreases as people age, and inequity exists between age groups. However, there was a considerable gap between High SDI countries and Low SDI countries. In 2019, the QCI in high SDI countries was >90 for all ages. QCI remained >75 for ages <49 in Low-SDI countries and then decreased to its dew point at less than 75 for ages 50–54 (Fig 3).

## Gender disparity ratio

GDR was calculated by dividing the age-standardized QCI of women by that of men: GDR closer to one indicated better equality. In 2019, the overall average of age-standardized GDR was 1.01 globally, which indicated that QCI was similar among women and men. Of 204 countries, the GDR of 100 countries ranged from 0.97 to 1.03, which was considered equality among both genders. Fifteen countries had GDR equal to one. Guinea-Bissau with GDR equal to 1.3 had the worst quality of care for men comparing with women in 2019. On the other hand, Honduras, Kiribati, and Pakistan, with GDR equal to 0.15, 0.71, and 0.78, had the highest gender disparity in favor of men in 2019 (S6 Table).

The gap between QCI of women and men has become narrower from 1990 to 2019. As opposed to High SDI countries, there was a significant gap between the GDR of older age groups and younger age groups in Low-SDI countries in 2019. In younger age groups, women had higher QCI than men, which was then conversed for middle age groups in favor of men, and then again in favor of women among older adults (Fig 4).

## Discussion

The global age-standardized QCI improved from 72.6 in 1990 to 83.2 in 2019, and nearly all countries and territories increased their index during this time. Singapore, Turkey, Japan, the Republic of Korea, and the United States of America had the highest QCI in 2019. Although higher SDI countries generally had higher QCI, there was significant heterogeneity for countries at similar SDI. Notwithstanding, the gap between the highest and lowest QCI was not bridged in the last three decades.

High-income Asia pacific had the highest QCI, both in 1990 and 2019. Interestingly, the prevalence of gastric cancer was highest in the high-income Asia Pacific [31]. In this region, Singapore, Japan, and the Republic of Korea had the highest QCI in 1990 and 2019. As a gastric cancer prevalent country, Japan implemented various screening programs for PUD and gastric cancer [32], which have been most successful in decreasing its prevalence [31]. Over the past years, Japan has been following a screen and treat approach, especially for adolescents, since Helicobacter pylori are mainly acquired during childhood [33]. In contrast to the Republic of Korea, QCI in the Democratic People's Republic of Korea decreased in the last three decades, highlighting the wide gap in their economic development [34].

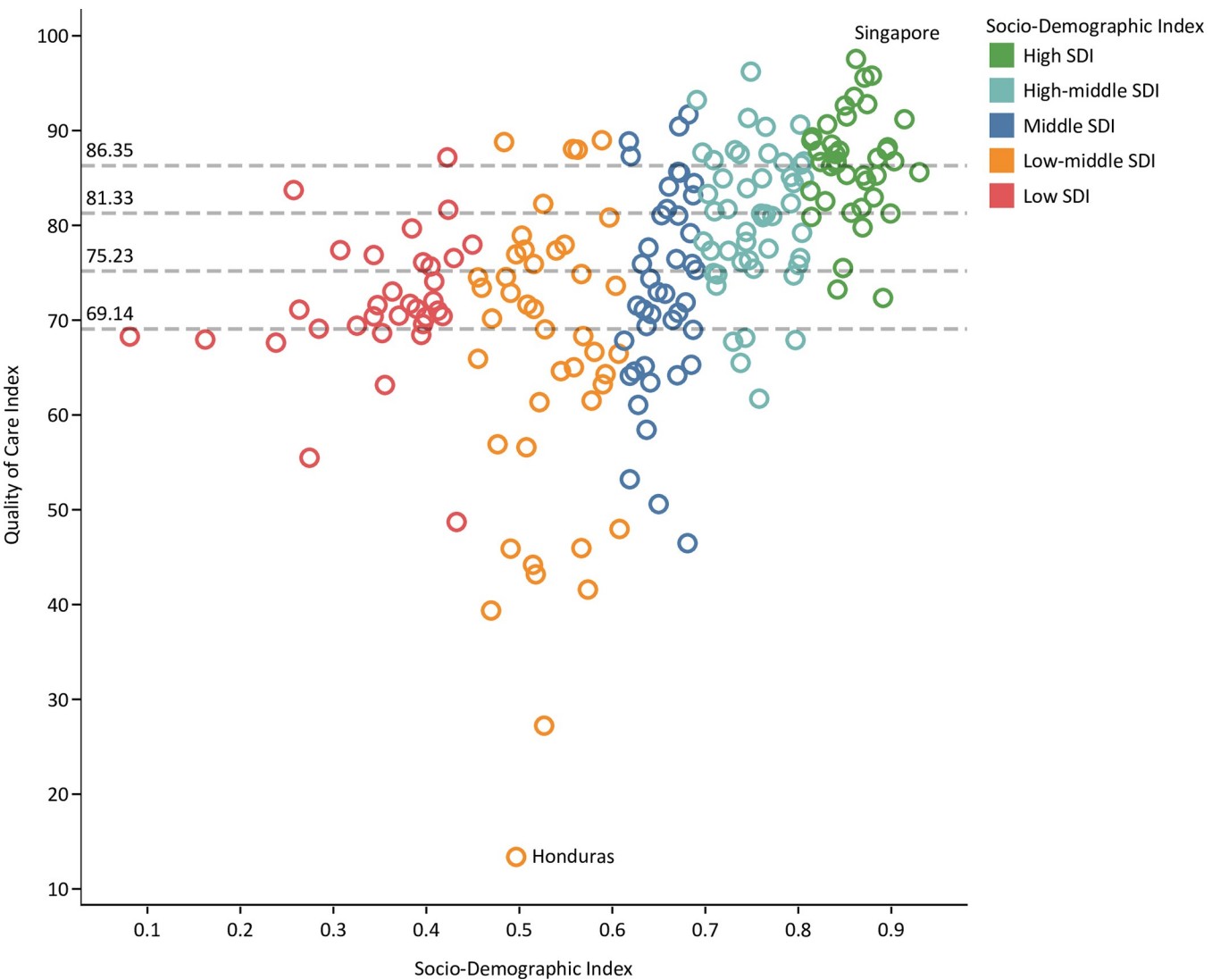

**Fig 2. Quality of care index for peptic ulcer disease by SDI score of countries in 2019.**

Although younger age groups had significantly higher QCI both in high and low SDI countries, the inequities among age groups decreased during 1990–2019. PUD in the elderly needs sustained attention since its presentation is insidious compared to younger patients; thus, diagnosis is delayed [35]. Older age is also associated with a higher incidence of Helicobacter pylori-associated and non-Helicobacter pylori-associated PUD due to the dramatic increase in the prescription of antithrombotic agents and non-steroidal anti-inflammatory drugs (NSAIDs) [36]. The mortality of bleeding peptic ulcers remains higher among patients over 60 years of age, especially those over 80, despite previous advances in pharmacological and endoscopic treatment [37]. PUD diagnosis and treatment approaches could be incredibly costly for low SDI countries, which could become a predisposing factor for favoring resource allocation for younger age groups.

The global GDR was close to one, indicating that QCI was similar among women and men. In addition, the GDR of nearly half of the countries ranged from 0.97 to 1.03. Although GDR could bring some insights towards seeing the big picture, it could also become misleading. In

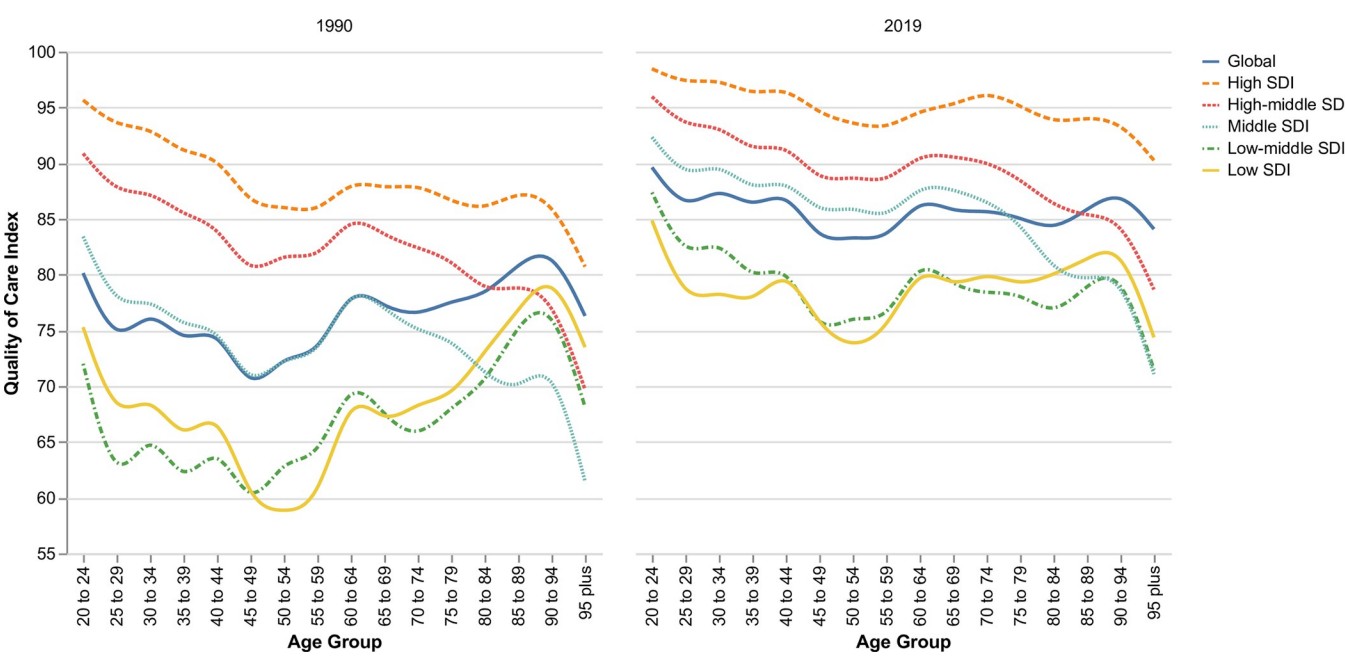

**Fig 3. Age pattern for quality of care index based on SDI in 1990 and 2019.**

case the QCI of both women and men is low, the ratio would seem satisfactory. Higher QCI among women could be due to increased PUD complications among men [38]. Nevertheless, the novel emerging evidence on the predisposing factors of complicated or non-complicated PUD would suggest that the witnessed inequities among age groups across various countries could be due to otherwise neglected factors such as the number of household members [39].

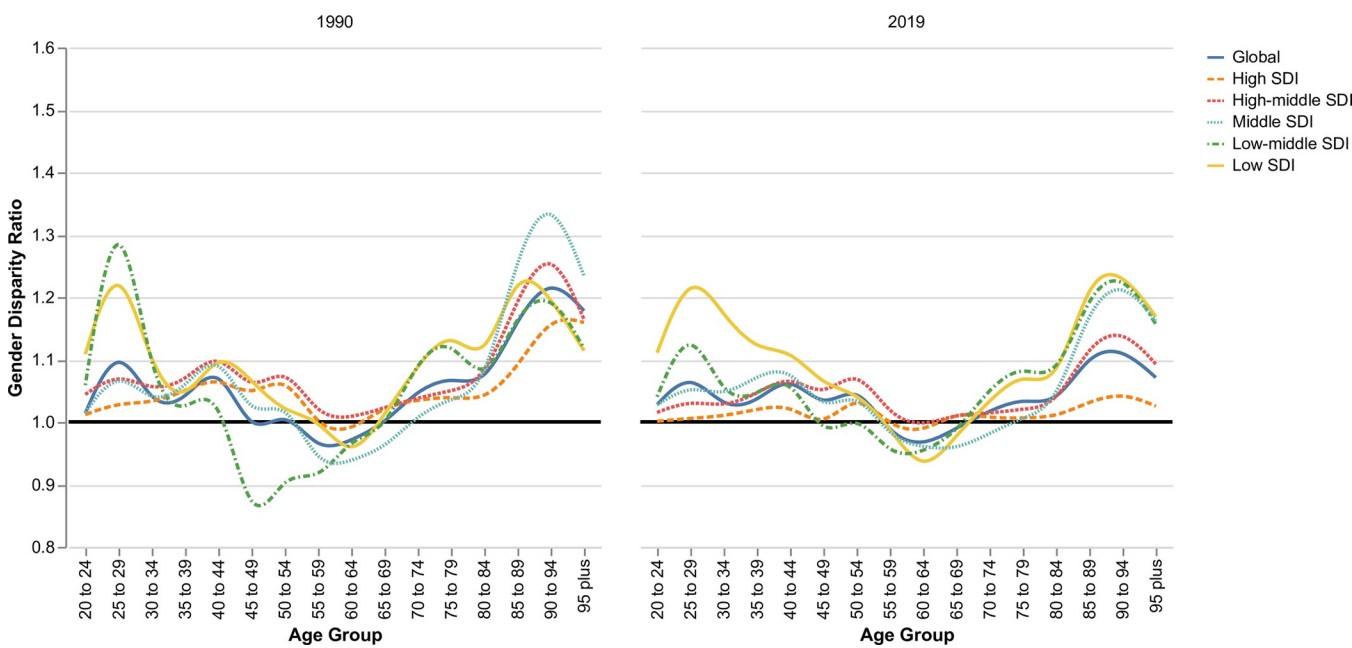

**Fig 4. Gender disparity ratio for quality of care index of peptic ulcer disease among age groups and SDI levels in 1990 and 2019.**

Thus, further research is required to enhance the current understanding of the existing inequities to help bridge the gaps among various groups, especially in low and low-middle income countries.

From 1990 to 2019, there has been a 31% reduction in the global age-standardized prevalence rate of PUD, 31% in age-standardized incidence rate, and 59% in age-standardized mortality rate. Other studies also showed a sharp decreasing trend in the prevalence, incidence, and mortality associated with PUD over the past 2–3 decades [2, 4]. The decline in the incidence and prevalence of PUD coincides with the decline in the prevalence of Helicobacter pylori. However, preventing factors like using medications that decrease gastric acid secretion and predisposing factors like smoking and the availability of non-steroidal, anti-inflammatory drugs need to be taken into account [40].

This study also highlights the considerable existing heterogeneity across the globe regarding the changes in prevalence, incidence, and mortality of PUD. While the age-standardized mortality rate due to PUD has decreased by 69% in high-SDI countries, low and low-middle SDI countries remain to have the highest age-standardized PUD mortality rate despite previous endeavors. The reasons for the witnessed heterogeneity could be larger family size, low socioeconomic status, overcrowding, poor sanitation, having an infected sibling, growth retardation, and nutritional deficiencies, particularly iron-deficiency anemia in low SDI countries [41, 42]. The witnessed gaps and inequities call for concerted efforts to lessen the burden of PUD in areas with limited resources. Given the level of technology, expenses, and the expertise required for complications management of PUD, its early detection and prevention seem feasible and cost-effective [43]. However, further research is required to determine the most suitable strategies for early detection, treatment, and follow-up based on available resources.

## Strengths and limitations

QCI combined mortality to incidence ratio [44], DALYs to prevalence ratio [45], prevalence to incidence ratio [46], and YLLs to YLDs ratio [47] into a single index, aiming to demonstrate the quality of care among countries [9–11, 28, 47–49]. Although QCI does not reflect all the aspects of the quality of services among healthcare systems, it could be used as a proxy for comparing various countries. The review of the existing literature also confirmed that the current situation of PUD on a global scale and its time trends during 1990–2019 mainly were consistent with the picture, as shown via QCI [3, 4, 27, 50]. In addition, it showed an acceptable correlation with the HAQ Index for PUD. The advantage of QCI is that, unlike HAQ Index, it could be used to present inequities among both sexes, age groups, and in all the seven GBD super-regions and 21 regions. QCI combined mortality to incidence ratio [44], DALYs to prevalence ratio [45], prevalence to incidence ratio [46], and YLLs to YLDs ratio [47], into a single index, aiming to demonstrate the quality of care among countries [9–11, 28, 47–49]. Although QCI does not reflect all the aspects of the quality of services among healthcare systems, it could be used as a proxy for comparing various countries. The review of the existing literature also confirmed that the current situation of PUD on a global scale and its time trends during 1990–2019 was mainly were mostly consistent with the picture, as shown via QCI [3, 4, 27, 50]. In addition, it showed an acceptable correlation with the HAQ Index for PUD. The advantage of QCI is that, unlike HAQ Index, it could be used to present inequities among both sexes, age groups, and in all the seven GBD super-regions and 21 regions.

Moreover, we used the estimates of GBD 2019, while the latest published HAQ Index used the data of GBD 2016 [28]. However, QCI does not currently capture subnational inequalities, and future efforts with a better geospatial resolution on sub-national levels need to be prioritized. Despite using as many data sources as possible, the GBD study includes estimations

based on the predictive covariates in locations with scarce data sources. Since the data of this study were derived from the GBD study, the limitations experienced in GBD estimations also apply to this study [51]. Among risk factors of PUD, smoking was the only risk factor with sufficient data to be modeled in the GBD study 2019. In this study, PCA was used to generate QCI, a mathematical method [52]. GBD utilizes statistical methods to estimate the burden measures, i.e. DALYs, YLLs, YLDs, death, incidences, and prevalence. Nevertheless, PCA did not allow us to present any uncertainty values; thus, the study does not report confidence or uncertainty intervals. Despite its limitations, the current study could show the big picture of the current inequities regarding PUD management based on geographical distribution, sex, and age groups to empower policymakers in making well-informed decisions. Moreover, we used the estimates of GBD 2019, while the latest published HAQ Index used the data of GBD 2016 [28]. However, QCI does not currently capture subnational inequalities, and future efforts with a better geospatial resolution on sub-national levels need to be prioritized. Despite using as many data sources as possible, the GBD study includes estimations based on the predictive covariates in locations with scarce data sources. Since the data of this study were derived from the GBD study, the limitations experienced in GBD estimations also apply to this study [51]. Among risk factors of PUD, smoking was the only risk factor with sufficient data to be modeled in the GBD study 2019. Unfortunately In this study, PCA was used to generate QCI, a mathematical method [52]. GBD utilizes statistical methods to estimate the burden measures, i.e. DALYs, YLLs, YLDs, death, incidences, and prevalence. Nevertheless, PCA did not allow us to present any uncertainty values; thus, the study does not report confidence or uncertainty intervals limited statistical and computational resources hindered calculating the confidence interval for all estimated indices and significant values in this study. Despite its limitations, the current study could be used for showing show the big picture of the current inequities regarding PUD management based on geographical distribution, sex, and age groups to empower policymakers towards in making well-informed decisions.

## Conclusions

QCI of PUD improved dramatically during 1990–2019 worldwide. There are still significant heterogeneities among countries on different and similar SDI levels. Given the inequities among various age groups and both sexes, there is still room for improvement for QCI and bridging the inequity gaps. The prosperous countries in this regard need to be introduced as role models, and their plans be implemented towards a sustained reduction of age-standardized death rates from the causes amenable to personal health care.

## Supporting information

**S1 Table. Data sources of GBD study 2019 for peptic ulcer disease.**
(XLSX)

**S2 Table. Mixed-effect regression analysis to assess the validation of QCI for PUD.**
(DOCX)

**S3 Table. Prevalence, incidence, and mortality of peptic ulcer disease among countries in 1990 and 2019.**
(DOCX)

**S4 Table. Changes in prevalence, incidence, and mortality of peptic ulcer disease among countries from 1990 to 2019.**
(DOCX)

**S5 Table. Age-standardized quality of care index and its levels of countries based on GBD super regions, regions, and SDI in 1990 and 2019.** * Locations were sorted by countries' QCI in 2019. (XLSX)

**S6 Table. Gender disparity ratio of countries in 1990 and 2019.** * Locations were sorted by countries' GDR in 2019. (XLSX)

## Acknowledgments

The authors sincerely thank all the collaborators who contributed to this study at Non-Communicable Diseases Research Center (NCDRC) and Endocrinology and Metabolism Research Institute (EMRI) at Tehran University of Medical Sciences, Tehran, Iran.

### Ethics approval statement

The study data were initially derived from the Global Burden of Disease study, Institute for Health Metrics and Evaluation, which abides by relevant guidelines and regulations. Data from the GBD study are freely available to researchers and policymakers [53]. Notably, the first author and the corresponding author were collaborators of the GBD study. This study was approved by the institutional review board of Endocrinology and Metabolism Research Institute at Tehran University of Medical Sciences (IR.TUMS.EMRI.REC.1400.016).

## Author Contributions

**Conceptualization:** Mohsen Abbasi-Kangevari, Naser Ahmadi, Nima Fattahi, Negar Rezaei, Mohammad-Reza Malekpour, Seyyed-Hadi Ghamari, Sahar Saeedi Moghaddam, Sina Azadnajafabad, Zahra Esfahani, Bagher Larijani, Farshad Farzadfar.

**Data curation:** Sahar Saeedi Moghaddam.

**Formal analysis:** Shahin Roshani.

**Investigation:** Mohsen Abbasi-Kangevari, Nima Fattahi, Negar Rezaei, Mohammad-Reza Malekpour, Seyyed-Hadi Ghamari, Sahar Saeedi Moghaddam, Sina Azadnajafabad, Zahra Esfahani, Ali-Asghar Kolahi, Shahin Roshani, Sahba Rezazadeh-Khadem, Fateme Gorgani, Seyyed Nima Naleini, Shohreh Naderimagham.

**Methodology:** Naser Ahmadi, Farshad Farzadfar.

**Resources:** Bagher Larijani, Farshad Farzadfar.

**Supervision:** Bagher Larijani, Farshad Farzadfar.

**Visualization:** Naser Ahmadi.

**Writing – original draft:** Mohsen Abbasi-Kangevari.

**Writing – review & editing:** Naser Ahmadi, Nima Fattahi, Negar Rezaei, Mohammad-Reza Malekpour, Seyyed-Hadi Ghamari, Sahar Saeedi Moghaddam, Sina Azadnajafabad, Zahra Esfahani, Shahin Roshani, Sahba Rezazadeh-Khadem, Fateme Gorgani, Seyyed Nima Naleini, Shohreh Naderimagham, Bagher Larijani, Farshad Farzadfar.

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
