## [Decision Letter · Decision Letter 0]

11 May 2022

PONE-D-22-00730Quality of care of peptic ulcer disease worldwide: a systematic analysis for the global burden of disease study 1990-2019PLOS ONE

Dear Dr. Farzadfar,

Thank you for submitting your manuscript to PLOS ONE. After careful consideration, we feel that it has merit but does not fully meet PLOS ONE’s publication criteria as it currently stands. Therefore, we invite you to submit a revised version of the manuscript that addresses the points raised during the review process.

We look forward to receiving your revised manuscript.

Kind regards,

Dinh-Toi Chu, PhD

Academic Editor

PLOS ONE

Journal Requirements: 

2. We note that Figure (1)  in your submission contain [map/satellite] images which may be copyrighted. All PLOS content is published under the Creative Commons Attribution License (CC BY 4.0), which means that the manuscript, images, and Supporting Information files will be freely available online, and any third party is permitted to access, download, copy, distribute, and use these materials in any way, even commercially, with proper attribution. For these reasons, we cannot publish previously copyrighted maps or satellite images created using proprietary data, such as Google software (Google Maps, Street View, and Earth). For more information, see our copyright guidelines: http://journals.plos.org/plosone/s/licenses-and-copyright.

1. You may seek permission from the original copyright holder of Figure (1) to publish the content specifically under the CC BY 4.0 license.  

Reviewer #1: The paper tackles a public health issue using measures such as the socio demographical index, in addition to using a proxy for quality of care, as well as the gender disparity ratio to assess the levels of inequities and quality of care for peptic ulcer disease worldwide among different demographical aspects.

The paper is simple and its writing style is comprehensible for the non specialized of readers. It uses easy to understand figures and graphs. The conclusion is supported by valid data.

This kind of analysis is useful in aiding policy makers in deciding on interventions to combat peptic ulcer disease at the root cause, because the study takes into consideration sociodemographic determinants of health, which in turn affect health outcomes and quality of care.

Reviewer #2: 1. All acronyms should be defined at the first mention of each of the acronyms in the manuscript.

2. What is NSAIDs?

3. What are the criteria for the classifications of GBD super regions?

4. Please remove the list of abbreviations at the end of the manuscript.

Reviewer #3: This study has been conducted using the data derived from the Global Burden of Disease study 19990 – 2019. The objective of this study was to compare the health-system quality-of-care and inequities for Peptic Ulcer Disease (PUD) among age groups and sexes worldwide. The authors have concluded that the Quality of Care Index of PUD improved dramatically during 1990-2019 worldwide. There are still significant heterogeneities among countries both on different and on similar SDI-levels.

The rationale for the study is clear and valid. The authors have used a technically suitable protocol and a feasible methodology to effectively achieve the study's aims.

However, I have the following concerns about their methodology.

1.) QCI is a relatively new index attempting to estimate the health care quality at the population level. The index was developed using a linear combination of mortality to incidence, DALY to prevalence, prevalence to incidence, and YLL to YLD ratios estimated using an acceptable statistical method. However, the authors should elaborate on interpreting the QCI relevant to the subject matter under consideration. To me, the QCI is too composite.

2.) The rationale of selecting or dividing the QCI into different levels (Level 1 to 5) is not clearly explained in terms of their relative importance. For example, the difference between levels 1 and 5 when considering the service delivery outcomes at the operational level is challenging to comprehend. How a country should move from level 1 to 5 is not clear.

3.) The major drawback was that the estimates lacked the confidence intervals to compare the different levels defined. The study's authors have highlighted this drawback, but it is better to get expert statistical advice to overcome this fundamental limitation.

4) Few spelling mistakes were noted.

---

## [Author Response · Author response to Decision Letter 0]

28 May 2022

Dear Editor in Chief

Please find enclosed the revised version of our manuscript titled "Quality of care of peptic ulcer disease worldwide: a systematic analysis for the global burden of disease study 1990-2019", which we would like to submit for publication in PLOS ONE.

Please find enclosed our reply to the editors and reviewer's comments. We have appreciated the encouraging, fair and constructive comments of the editors and reviewers. We revised the manuscript according to the suggestions and enclosed a highlighted version of the revised manuscript.

We feel that the changes made according to the comments have improved the quality of the manuscript, and we would be happy if it now meets the criteria for publication in the journal. 

Best regards; 

Farshad Farzadfar

(On behalf of myself and my co-authors)

 

Editor comments

Editor

Authors

Amended.

Editor

We note that Figure (1) in your submission contain [map/satellite] images which may be copyrighted. All PLOS content is published under the Creative Commons Attribution License (CC BY 4.0), which means that the manuscript, images, and Supporting Information files will be freely available online, and any third party is permitted to access, download, copy, distribute, and use these materials in any way, even commercially, with proper attribution. For these reasons, we cannot publish previously copyrighted maps or satellite images created using proprietary data, such as Google software (Google Maps, Street View, and Earth). For more information, see our copyright guidelines: http://journals.plos.org/plosone/s/licenses-and-copyright. 

 1. You may seek permission from the original copyright holder of Figure (1) to publish the content specifically under the CC BY 4.0 license. 

Authors

Kindly be informed that the maps were generated using free, open-source map data of the Natural Earth public domain (naturalearthdata.com) via Python programming language. This was also included in the methods section of the manuscript. It now reads:

The maps were generated using free open-source map data of Natural Earth public domain (naturalearthdata.com) via Python programming language.

Editor

We note that you have indicated that data from this study are available upon request. PLOS only allows data to be available upon request if there are legal or ethical restrictions on sharing data publicly. For information on unacceptable data access restrictions, please see http://journals.plos.org/plosone/s/data-availability#loc-unacceptable-data-access-restrictions. 

Authors

Thank you for your meticulous comment. Regarding the data availability, we have previously published the study protocol to enable other researchers to reproduce this work. Moreover, the data used in this study are available from the Global Burden of Disease Results Tool, made public by Institute for Health Metrics and Evaluation. The authors confirm they had no special access or privileges to the data that other researchers would not have. We revised the data source sub-heading in the methods section, which now reads:

The study data were derived from GBD 2019, conducted by IHME. GBD 2019 included 204 countries and territories from 1990 to 2019 and a systematic analysis of 286 causes of death, 369 diseases and injuries, and 87 risk factors in 204 countries and territories [1,2]. GBD classified countries and territories into 21 regions based on epidemiological homogeneity and geographical contiguity [3]. The regions were also grouped into seven super-regions based on the cause of death patterns [4]. The seven super-regions are High income; Latin America & Caribbean; Sub-Saharan Africa; North Africa & Middle East; Southeast Asia, East Asia & Oceania; South Asia; Central Europe, Eastern Europe & Central Asia. 

The study protocol and codes used in this study are available from (https://dx.doi.org/10.17504/protocols.io.bprjmm4n) [5]. The data used in this work are available from Global Burden of Disease Results Tool (http://ghdx.healthdata.org/gbd-results-tool) [6], made public by Institute for Health Metrics and Evaluation. The data of PUD were extracted from GBD 2019: GBD code: B.4.2.1, International Statistical Classification of Diseases and Related Health Problems 10th Revision, World Health Organization version 10 (ICD-10) code: K-25 to K28.9 [7]. Data sources used to provide estimates in GBD 2019 are presented in Supplementary Table 1. In terms of the development status, countries were categorized using the GBD Socio-Demographic Index (SDI) [8]. 

 

Reviewer 1

Reviewer

Reviewer #1: The paper tackles a public health issue using measures such as the socio demographical index, in addition to using a proxy for quality of care, as well as the gender disparity ratio to assess the levels of inequities and quality of care for peptic ulcer disease worldwide among different demographical aspects.

The paper is simple and its writing style is comprehensible for the non specialized of readers. It uses easy to understand figures and graphs. The conclusion is supported by valid data.

This kind of analysis is useful in aiding policy makers in deciding on interventions to combat peptic ulcer disease at the root cause, because the study takes into consideration sociodemographic determinants of health, which in turn affect health outcomes and quality of care.

Authors

The authors appreciate the encouraging comments of the reviewer.

 

Reviewer 2

Reviewer

1. All acronyms should be defined at the first mention of each of the acronyms in the manuscript.

Authors

The authors appreciate the fair and constructive comment of the reviewer. Amended. 

Reviewer

2. What is NSAIDs?

Authors

Thank you for your comment. We double-checked the manuscript and realized that the abbreviation "NSAID" (non-steroidal anti-inflammatory drugs) was not defined in the text, which was then included in the text. 

Reviewer

3. What are the criteria for the classifications of GBD super regions?

Authors

GBD 2019 included 204 countries and territories from 1990 to 2019 and a systematic analysis of 286 causes of death, 369 diseases and injuries, and 87 risk factors in 204 countries and territories [1,2]. GBD classified countries and territories into 21 regions based on epidemiological homogeneity and geographical contiguity [3]. The regions were also grouped into seven super-regions based on the cause of death patterns [4]. The seven super-regions are High income; Latin America & Caribbean; Sub-Saharan Africa; North Africa & Middle East; Southeast Asia, East Asia & Oceania; South Asia; Central Europe, Eastern Europe & Central Asia. 

The data source section of the methods was revised and now reads: 

The study data were derived from GBD 2019, conducted by IHME. GBD 2019 included 204 countries and territories from 1990 to 2019 and a systematic analysis of 286 causes of death, 369 diseases and injuries, and 87 risk factors in 204 countries and territories [1,2]. GBD classified countries and territories into 21 regions based on epidemiological homogeneity and geographical contiguity [3]. The regions were also grouped into seven super-regions based on the cause of death patterns [4]. The seven super-regions are High income; Latin America & Caribbean; Sub-Saharan Africa; North Africa & Middle East; Southeast Asia, East Asia & Oceania; South Asia; Central Europe, Eastern Europe & Central Asia. The data of PUD was extracted from GBD 2019: GBD code: B.4.2.1, International Statistical Classification of Diseases and Related Health Problems 10th Revision, World Health Organization version 10 (ICD-10) code: K-25 to K28.9 [7]. Data sources used to provide estimates in GBD 2019 are presented in Supplementary Table 1. In terms of the development status, countries were categorized using the GBD Socio-Demographic Index (SDI) [8].

Reviewer

4. Please remove the list of abbreviations at the end of the manuscript.

Authors

Thank you for your comment. Amended.

Reviewer 3

Reviewer

This study has been conducted using the data derived from the Global Burden of Disease study 19990 – 2019. The objective of this study was to compare the health-system quality-of-care and inequities for Peptic Ulcer Disease (PUD) among age groups and sexes worldwide. The authors have concluded that the Quality of Care Index of PUD improved dramatically during 1990-2019 worldwide. There are still significant heterogeneities among countries both on different and on similar SDI-levels.

The rationale for the study is clear and valid. The authors have used a technically suitable protocol and a feasible methodology to effectively achieve the study's aims.

However, I have the following concerns about their methodology.

Authors

The authors would like to thank the reviewer for the encouraging opinion. They also appreciate the fair and constructive comments of the reviewer.

Reviewer

1.) QCI is a relatively new index attempting to estimate the health care quality at the population level. The index was developed using a linear combination of mortality to incidence, DALY to prevalence, prevalence to incidence, and YLL to YLD ratios estimated using an acceptable statistical method. However, the authors should elaborate on interpreting the QCI relevant to the subject matter under consideration. To me, the QCI is too composite.

Authors

The authors appreciate the fair and constructive comment of the reviewer. We reviewed the methods section and agree that further details need to be included. The quality of care index section in the methods now reads:

To determine QCI for PUD, the following indices were defined as follows: 

Mortality to incidence ratio of PUD=(# Age-standardizaed PUD mortality)/(# Age-standardizaed PUD Incidence)

The mortality to incidence ratio of PUD indicates that with a stable PUD incidence in regions, higher mortality values could represent worse care provided to these patients.

DALY to prevalence ratio of PUD=(# Age-standardizaed DALY of PUD )/(# Age-standardizaed PUD prevalence )

DALY to prevalence ratio of PUD indicates that with a similar prevalence of PUD in regions, higher DALY could represent worse care quality.

 Prevalence to incidence ratio of PUD=(# Age-standardizaed PUD prevalence )/(# Age-standardizaed PUD incidence)

The prevalence to incidence ratio of PUD indicates that in regions with similar PUD incidence, higher prevalence could represent better PUD management to avert mortality.

YLL to YLD ratio of PUD=(# Age-standardizaed YLL of PUD )/(# Age-standardizaed YLD of PUD)

YLL to YLD ratio of PUD could reflect the quality of healthcare in a region, as poor health quality provided for PUD in a region would result in higher YLLs and fewer YLDs. In other words, patients would live less than the life expectancy of the region).

Moreover, the section describing the Principal Components Analysis was revised, which now reads:

PCA would allow us to reduce the number of variables in a large set of correlated variables to a smaller number of variables that collectively explain most of the variance in the original set [9]. The first component of PCA was made up of a linear combination of the four abovementioned ratios, including mortality to incidence, DALY to prevalence, prevalence to incidence, and YLL to YLD, containing the most significant amount of information about these variables and is referred to as QCI in this study. QCI ranged from 0 to 100, with 100 indicating the best quality of care [10–14]. 

Reviewer

2.) The rationale of selecting or dividing the QCI into different levels (Level 1 to 5) is not clearly explained in terms of their relative importance. For example, the difference between levels 1 and 5 when considering the service delivery outcomes at the operational level is challenging to comprehend. How a country should move from level 1 to 5 is not clear.

Authors

Thank you for your meticulous comment. For enhanced interpretation and comparison of countries, QCI was categorized as five levels in 2019 based on quintiles, where Level 1 (the first quintile) indicated the highest index and Level 5 (the fifth quintile) the lowest: Level 5 included QCI ≤69.14, Level 4 QCI>69.14 to 75.23, Level 3 QCI>75.23 to 81.33, Level 2 QCI>81.33 to 86.35, and Level 1 QCI≥86.35.

We revised the corresponding section in the methods, which now reads:

PCA was performed using R software version 3.5.2. For enhanced interpretation and comparison of countries, QCI was categorized as five levels in 2019 based on quintiles, where Level 1 (the first quintile) indicated the highest index and Level 5 (the fifth quintile) the lowest: Level 5 included QCI ≤69.14, Level 4 QCI>69.14 to 75.23, Level 3 QCI>75.23 to 81.33, Level 2 QCI>81.33 to 86.35, and Level 1 QCI≥86.35

We also revised the last paragraph of the discussion on the witnessed heterogeneities across countries in terms of PUD:

This study also highlights the considerable existing heterogeneity across the globe regarding the changes in prevalence, incidence, and mortality of PUD. While the age-standardized mortality rate due to PUD has decreased by 69% in high-SDI countries, low and low-middle SDI countries remain to have the highest age-standardized PUD mortality rate despite previous endeavors. The reasons for the witnessed heterogeneity could be larger family size, low socioeconomic status, overcrowding, poor sanitation, having an infected sibling, growth retardation, and nutritional deficiencies, particularly iron-deficiency anemia in low SDI countries [15,16]. The witnessed gaps and inequities call for concerted efforts to lessen the burden of PUD in areas with limited resources. Given the level of technology, expenses, and the expertise required for complications management of PUD, its early detection and prevention seem feasible and cost-effective [17]. However, further research is required to determine the most suitable strategies for early detection, treatment, and follow-up based on available resources. 

Reviewer

3.) The major drawback was that the estimates lacked the confidence intervals to compare the different levels defined. The study's authors have highlighted this drawback, but it is better to get expert statistical advice to overcome this fundamental limitation.

Authors

Thank you for your meticulous comment. Kindly note that in this study, PCA was used to generate QCI, which is a mathematical method [18]. GBD utilizes statistical methods to estimate the burden measures, i.e. DALYs, YLLs, YLDs, death, incidences, and prevalence. Nevertheless, PCA did not allow us to present any uncertainty values; thus, the study does not report confidence or uncertainty intervals.

We revised the strengths and limitations section of the manuscript accordingly, which now reads:

QCI combined mortality to incidence ratio [19], DALYs to prevalence ratio [20], prevalence to incidence ratio [21], and YLLs to YLDs ratio [22] into a single index, aiming to demonstrate the quality of care among countries [11–13,22–25]. Although QCI does not reflect all the aspects of the quality of services among healthcare systems, it could be used as a proxy for comparing various countries. The review of the existing literature also confirmed that the current situation of PUD on a global scale and its time trends during 1990-2019 mainly were consistent with the picture, as shown via QCI [26–29]. In addition, it showed an acceptable correlation with the HAQ Index for PUD. The advantage of QCI is that, unlike HAQ Index, it could be used to present inequities among both sexes, age groups, and in all the seven GBD super-regions and 21 regions.

Moreover, we used the estimates of GBD 2019, while the latest published HAQ Index used the data of GBD 2016 [23]. However, QCI does not currently capture subnational inequalities, and future efforts with a better geospatial resolution on sub-national levels need to be prioritized. Despite using as many data sources as possible, the GBD study includes estimations based on the predictive covariates in locations with scarce data sources. Since the data of this study were derived from the GBD study, the limitations experienced in GBD estimations also apply to this study [30]. Among risk factors of PUD, smoking was the only risk factor with sufficient data to be modeled in the GBD study 2019. In this study, PCA was used to generate QCI, which is a mathematical method [18]. GBD utilizes statistical methods to estimate the burden measures, i.e. DALYs, YLLs, YLDs, death, incidences, and prevalence. Nevertheless, PCA did not allow us to present any uncertainty values; thus, the study does not report confidence or uncertainty intervals. Despite its limitations, the current study could show the big picture of the current inequities regarding PUD management based on geographical distribution, sex, and age groups to empower policymakers in making well-informed decisions.

Reviewer

4) Few spelling mistakes were noted.

Authors

The authors appreciate the reviewer's opinion. We asked a bilingual native English professor at our institution to thoroughly review and revise the manuscript regarding possible grammar and syntax errors to ensure enhanced readability. In addition, we used "Grammarly", a cross-platform cloud-based writing assistant that reviews spelling, grammar, punctuation, clarity, engagement, and delivery mistakes.

 

References

1. Abbafati C, Machado DB, Cislaghi B, Salman OM, Karanikolos M, McKee M, et al. Global burden of 369 diseases and injuries in 204 countries and territories, 1990–2019: a systematic analysis for the Global Burden of Disease Study 2019. Lancet. 2020;396: 1204–1222. doi:10.1016/S0140-6736(20)30925-9

2. Murray CJL, Aravkin AY, Zheng P, Abbafati C, Abbas KM, Abbasi-Kangevari M, et al. Global burden of 87 risk factors in 204 countries and territories, 1990–2019: a systematic analysis for the Global Burden of Disease Study 2019. Lancet. 2020;396: 1223–1249. doi:10.1016/S0140-6736(20)30752-2

3. Murray CJL, Ezzati M, Flaxman AD, Lim S, Lozano R, Michaud C, et al. GBD 2010: design, definitions, and metrics. Lancet (London, England). 2012;380: 2063–2066. doi:10.1016/S0140-6736(12)61899-6

4. Frequently Asked Questions | Institute for Health Metrics and Evaluation. [cited 16 May 2022]. Available: https://www.healthdata.org/gbd/faq

5. Mohammadi E, Ghasemi E, Moghaddam SS, Yoosefi M, Roshani S, Ahmadi N, et al. Quality of Care Index (QCI). protocols.io. 2020. 

6. VizHub - GBD Results. [cited 22 May 2022]. Available: https://vizhub.healthdata.org/gbd-results/

7. Global, regional, and national incidence, prevalence, and years lived with disability for 354 diseases and injuries for 195 countries and territories, 1990-2017: a systematic analysis for the Global Burden of Disease Study 2017. Lancet (London, England). 2018;392: 1789–1858. doi:10.1016/S0140-6736(18)32279-7

8. Mokdad AH, Mensah GA, Krish V, Glenn SD, Miller-Petrie MK, Lopez AD, et al. Global, regional, national, and subnational big data to inform health equity research: Persp ectives from the global burden of disease study 2017. Ethn Dis. 2019;29: 159–172. doi:10.18865/ed.29.S1.159

9. Rencher AC, Schimek MG. Methods of multivariate analysis. Comput Stat. 1997;12: 422. 

10. Quality of Care Index (QCI). [cited 17 Mar 2021]. Available: https://www.protocols.io/view/quality-of-care-index-qci-bprjmm4n.html

11. Azadnajafabad S, Saeedi Moghaddam S, Mohammadi E, Rezaei N, Ghasemi E, Fattahi N, et al. Global, regional, and national burden and quality of care index (QCI) of thyroid cancer: A systematic analysis of the Global Burden of Disease Study 1990–2017. Cancer Med. 2021;00: cam4.3823. doi:10.1002/cam4.3823

12. Mohammadi E, Ghasemi E, Azadnajafabad S, Rezaei N, Saeedi Moghaddam S, Ebrahimi Meimand S, et al. A global, regional, and national survey on burden and Quality of Care Index (QCI) of brain and other central nervous system cancers; global burden of disease systematic analysis 1990-2017. Huang T, editor. PLoS One. 2021;16: e0247120. doi:10.1371/journal.pone.0247120

13. Keykhaei M, Masinaei M, Mohammadi E, Azadnajafabad S, Rezaei N, Saeedi Moghaddam S, et al. A global, regional, and national survey on burden and Quality of Care Index (QCI) of hematologic malignancies; global burden of disease systematic analysis 1990–2017. Exp Hematol Oncol. 2021;10: 11. doi:10.1186/s40164-021-00198-2

14. Aryannejad A, Tabary M, Ebrahimi N, Mohammadi E, Fattahi N, Roshani S, et al. Global, regional, and national survey on the burden and quality of care of pancreatic cancer; a systematic analysis for the Global Burden of Disease study 1990–2017. Pancreatology. 2021. doi:10.1016/J.PAN.2021.09.002

15. Abdul Rahim NR, Benson J, Grocke K, Vather D, Zimmerman J, Moody T, et al. Prevalence of Helicobacter pylori infection in newly arrived refugees attending the Migrant Health Service, South Australia. Helicobacter. 2017;22: e12360. doi:10.1111/hel.12360

16. Mungazi SG, Chihaka OB, Muguti GI. Prevalence of Helicobacter pylori in asymptomatic patients at surgical outpatient department: Harare hospitals. Ann Med Surg. 2018;35: 153–157. doi:10.1016/j.amsu.2018.09.040

17. Lansdorp-Vogelaar I, Meester RGS, Laszkowska M, Escudero FA, Ward ZJ, Yeh JM. Cost-effectiveness of prevention and early detection of gastric cancer in Western countries. Best Pract Res Clin Gastroenterol. 2021;50–51: 101735. doi:10.1016/J.BPG.2021.101735

18. Johnson RA, Wichern DW. Applied multivariate statistical analysis. Pearson London, UK:; 2014. 

19. Choi E, Lee S, Nhung BC, Suh M, Park B, Jun JK, et al. Cancer mortality-to-incidence ratio as an indicator of cancer management outcomes in Organization for Economic Cooperation and Development countries. Epidemiol Health. 2017;39: e2017006. doi:10.4178/epih.e2017006

20. Chang AY, Skirbekk VF, Tyrovolas S, Kassebaum NJ, Dieleman JL. Measuring population ageing: an analysis of the Global Burden of Disease Study 2017. Lancet Public Heal. 2019;4: e159–e167. doi:10.1016/S2468-2667(19)30019-2

21. Han C, Zhao N, Gaslightwala A, Bala M. An epidemiological and healthcare utilisation study of rheumatoid arthritis in an adult population in the US. J Med Econ. 2007;10: 489–499. doi:10.3111/13696990701701745

22. Khalil I, El Bcheraoui C, Charara R, Moradi-Lakeh M, Afshin A, Kassebaum NJ, et al. Transport injuries and deaths in the Eastern Mediterranean Region: findings from the Global Burden of Disease 2015 Study. Int J Public Health. 2018;63: 187–198. doi:10.1007/s00038-017-0987-0

23. Measuring performance on the Healthcare Access and Quality Index for 195 countries and territories and selected subnational locations: a systematic analysis from the Global Burden of Disease Study 2016. Lancet (London, England). 2018;391: 2236–2271. doi:10.1016/S0140-6736(18)30994-2

24. Momtazmanesh S, Saeedi Moghaddam S, Malakan Rad E, Azadnajafabad S, Ebrahimi N, Mohammadi E, et al. Global, regional, and national burden and quality of care index of endocarditis: the global burden of disease study 1990–2019. Eur J Prev Cardiol. 2021. 

25. Nejad M, Ahmadi N, Mohammadi E, Shabani M, Sherafati A, Aryannejad A, et al. Global and regional burden and quality of care of non-rheumatic valvular heart diseases: a systematic analysis of Global Burden of Disease 1990–2017. Int J Qual Heal Care. 2022;34: mzac026. 

26. Sung JJY, Kuipers EJ, El-Serag HB. Systematic review: the global incidence and prevalence of peptic ulcer disease. Aliment Pharmacol Ther. 2009;29: 938–946. doi:10.1111/j.1365-2036.2009.03960.x

27. Kavitt RT, Lipowska AM, Anyane-Yeboa A, Gralnek IM. Diagnosis and Treatment of Peptic Ulcer Disease. Am J Med. 2019;132: 447–456. doi:10.1016/j.amjmed.2018.12.009

28. Barber RM, Fullman N, Sorensen RJD, Bollyky T, McKee M, Nolte E, et al. Healthcare access and quality index based on mortality from causes amenable to personal health care in 195 countries and territories, 1990-2015: A novel analysis from the global burden of disease study 2015. Lancet. 2017;390: 231–266. doi:10.1016/S0140-6736(17)30818-8

29. Fullman N, Yearwood J, Abay SM, Abbafati C, Abd-Allah F, Abdela J, et al. Measuring performance on the Healthcare Access and Quality Index for 195 countries and territories and selected subnational locations: A systematic analysis from the Global Burden of Disease Study 2016. Lancet. 2018;391: 2236–2271. doi:10.1016/S0140-6736(18)30994-2

30. Roth GA, Abate D, Abate KH, Abay SM, Abbafati C, Abbasi N, et al. Global, regional, and national age-sex-specific mortality for 282 causes of death in 195 countries and territories, 1980–2017: a systematic analysis for the Global Burden of Disease Study 2017. Lancet. 2018;392: 1736–1788. doi:10.1016/S0140-6736(18)32203-7

---

## [Decision Letter · Decision Letter 1]

28 Jun 2022

Quality of care of peptic ulcer disease worldwide: a systematic analysis for the global burden of disease study 1990-2019

PONE-D-22-00730R1

Dear Dr. Farzadfar,

We’re pleased to inform you that your manuscript has been judged scientifically suitable for publication and will be formally accepted for publication once it meets all outstanding technical requirements.

Kind regards,

Dinh-Toi Chu, PhD

Academic Editor

PLOS ONE

---

## [Editor Report · Acceptance letter]

21 Jul 2022

PONE-D-22-00730R1 

Quality of care of peptic ulcer disease worldwide: a systematic analysis for the global burden of disease study 1990-2019 

Dear Dr. Farzadfar:

I'm pleased to inform you that your manuscript has been deemed suitable for publication in PLOS ONE. Congratulations! Your manuscript is now with our production department. 

Kind regards, 

on behalf of

Dr. Dinh-Toi Chu 

Academic Editor

PLOS ONE